# Heavy Metals in Umbilical Cord Blood: Effects on Epigenetics and Child Development

**DOI:** 10.3390/cells13211775

**Published:** 2024-10-26

**Authors:** Sudipta Dutta, Douglas M. Ruden

**Affiliations:** 1Department of Veterinary Integrative Biosciences, College of Veterinary Medicine and Biomedical Sciences, Texas A&M University, College Station, TX 77843, USA; sdutta@cvm.tamu.edu; 2C. S. Mott Center for Human Health and Development, Department of Obstetrics and Gynecology, Institute of Environmental Health Sciences, Wayne State University, Detroit, MI 48202, USA

**Keywords:** heavy metals, developmental toxicology, umbilical cord blood, placenta

## Abstract

Heavy metals like arsenic, mercury, cadmium, and lead are harmful pollutants that can change how our genes are regulated without altering the DNA sequence, specifically through a process called DNA methylation (DNAm) at 5-methylcytosine, an epigenetic mark that we will focus on in this review. These changes in DNAm are most sensitive during pregnancy, a critical time for development when these modifications can affect how traits are expressed. Historically, most research on these environmental effects has focused on adults, but now there is more emphasis on studying the impacts during early development and childhood. The placenta acts as a protective barrier between the mother and the baby, and by examining it, scientists can identify changes in key genes that might affect long-term health. This review looks at how exposure to heavy metals during pregnancy can cause changes in the gene regulation by DNAm in newborns, as seen in their umbilical cord blood. These changes reflect the baby’s genetic state during pregnancy and can be influenced by the mother’s environment and genetics, as well as the baby’s own genetics.

## 1. Introduction

The exposure of pregnant women and young children to environmental pollutants is a significant public health concern [1]. Research in environmental health indicates that fetal development is the most vulnerable period to these exposures. This understanding supports the Developmental Origins of Health and Disease (DOHaD) hypothesis, which links maternal exposure during pregnancy to potential health risks and diseases later in life for the offspring [2].

Fetal programming refers to the process where the in utero environment can prepare a fetus for the expected post-delivery environment, significantly impacting lifelong health [3]. Epigenetic mechanisms, particularly DNAm, play a crucial role in mediating fetal programming [4,5], potentially altering health and disease trajectories throughout the offspring’s lifetime [4,6,7]. Maternal smoking during pregnancy and gestational age are closely linked with differential DNAm in offspring cord blood, as evidenced by cord blood epigenome-wide association studies (EWASs) [3].

This paper examines how heavy metal exposures alter DNAm in umbilical cord blood and reviews both the short-term and long-term consequences of this exposure. We first describe the structure of the placenta and then discuss how the placenta is a gatekeeper for maternal and fetal exposures. With this background information, we then focus on the short-term effects of heavy metal exposure on offspring. Finally, we elaborate on the long-term effects on children exposed to heavy metals. Finally, Section 6 presents future directions for research in this field to better understand the mechanistic basis of how heavy metals alter DNAm in umbilical cord blood and how these changes impact child outcomes.

While there are several reviews discussing how environmental exposures affect DNAm in the placenta and its link to negative health outcomes [8,9,10,11], this is not the focus of the current review. By concentrating on umbilical cord blood instead of the placenta, we can investigate the effects on the first fetal tissue exposed to heavy metals and associated cytokines after they have passed through the placenta.

## 2. The Structure of the Placenta and the Umbilical Cord

The placenta is a unique, temporary organ that consists of both maternal and fetal parts and is the first and largest organ to develop during pregnancy [12]. It handles nutrient exchange and waste removal for the fetus, supporting growth and development within the mother’s body [13]. The side of the placenta facing the fetus is called the chorionic plate, which connects to the fetus via the umbilical cord [12]. The maternal side, known as the basal plate, attaches to and anchors into the mother’s uterine endometrial lining [12].

The umbilical cord serves as the primary link between the fetus and the placenta during pregnancy (Figure 1). It begins to form during the third week of embryonic formation and is fully formed by the seventh week. Structurally, the umbilical cord includes the connecting stalk, vitelline duct, and umbilical vessels, all encased by the amniotic membrane [14]. These umbilical vessels allow the exchange of blood between the fetus and the placenta: the umbilical vein transports oxygen-rich blood and nutrients from the placenta to the fetus, while the umbilical arteries transport deoxygenated blood and waste from the fetus to the placenta [15]. By birth, the umbilical cord typically measures 50–60 cm in length and 2 cm in diameter, with up to 40 helical twists [16,17].

By the end of the fifth week of pregnancy, blood flow between the mother and the fetus is established through three umbilical vessels: two arteries and one vein. These vessels have tunica intima and tunica media layers but lack the tunica adventitia layer [18]. Instead, the role of the tunica adventitia is taken over by Wharton’s jelly, a mucoid connective tissue [19]. The umbilical vessels are embedded in this gelatinous tissue, which prevents compression and torsion, ensuring proper bidirectional blood transfer between the mother and fetus, which is crucial for fetal development [20].

Wharton’s jelly, a type of connective tissue, is rich in an extracellular matrix that includes glycosaminoglycans (mainly hyaluronic acid) and various types of collagen [21]. It also contains fibroblast-like connective tissue cells and mast cells [22,23]. The jelly is covered by a cord lining with an outer layer of umbilical epithelium. This epithelium consists of one or more layers of epithelial cells, likely originating from the amnion, that exhibit a cytokeratin pattern-like human epidermis and resemble the skin’s epidermal and dermal layers [22,24,25].

Umbilical cord blood (UCB) is the blood contained within the placenta and umbilical cord vessels after delivery [26]. UCB contains at least three types of stem cells: predominantly hematopoietic stem cells (HSCs), multipotent mesenchymal stem cells (MSCs) [27,28,29], and a smaller concentration of non-hematopoietic multipotent stem cells that express the SSEA-4 protein, a marker found on embryonic stem cells [30,31]. It is rich in MSCs that express CD105, CD73, CD90, Oct-4, Sox-2, and Nanog, and these cells can differentiate into osteogenic, adipogenic, chondrogenic, and other lineages [32].

The identification of stem cells in residual umbilical cord and placental blood in the 1980s marked a significant breakthrough in medical science, revealing a viable alternative to bone marrow for transplant procedures. This discovery triggered the establishment and rapid expansion of stem cell banks on a global scale, allowing for the storage of these valuable cells for future medical use. Over the past two decades, the U.S. alone has seen the collection and preservation of over 4 million cord blood samples, a testament to the growing recognition of its medical potential. Of these, more than 40,000 samples have already been thawed and successfully used in various transplant therapies and regenerative medicine treatments, highlighting the real-world impact of cord blood banking [33].

This success has not only validated the significance of cord blood banking but has also catalyzed interest in the preservation of cord tissue, which contains different types of stem cells with promising regenerative capabilities. The banking of cord tissue is now gaining traction as researchers and clinicians look to the future, anticipating advances in regenerative medicine and tissue engineering that may further enhance medical treatments and the field of personalized medicine [33].

## 3. The Placenta as a Gatekeeper of Maternal and Fetal Exposures

The environment in which a child develops significantly affects their disease susceptibility later in life [34]. A growing body of evidence indicates that exposure to environmental contaminants found in air, water, soil, consumer products, and household items plays a crucial role in adverse pregnancy outcomes [35]. Toxic metals have been detected in the urine, peripheral blood, and amniotic fluid of pregnant women [35]. Due to their high toxicity, arsenic (As), cadmium (Cd), chromium (Cr), lead (Pb), and mercury (Hg) are among the priority metals of major public health concern [36].

Humans are exposed to various toxic heavy metals daily. For instance, lead (Pb) exposure can occur through household paints, toys, and groundwater contaminated by Pb-containing plumbing. Chromium (Cr) exposure comes from combustion processes and metal industries. Cadmium (Cd) can be ingested through drinking water contaminated by soil pollution, industrial activity, waste combustion, or smoking. Arsenic (As) exposure typically occurs through drinking water, and mercury (Hg) exposure can result from coal combustion, mining, consuming fish, dental amalgam, and thermometer use [37,38,39]. Heavy metals have very low excretion efficiency, allowing them to accumulate in the body and exert neurotoxic or immunotoxic effects, especially during early development [40].

The placenta acts as a “gatekeeper” between maternal and fetal exposures [35]. It can either fully filter or reduce fetal exposure to various toxicants, or, in some cases, allow certain metals to reach levels in cord blood that are higher than those detected in maternal samples [41,42] (Figure 2). This review focuses on heavy metal exposures, their effects on DNA methylation (DNAm) in umbilical cord blood (UCB), and the implications for the short-term and long-term health of the offspring.

## 4. Heavy Metal Exposures and Umbilical Cord Blood DNA Methylation

The harmful effects of heavy metals on human health stem from their ability to interfere with a range of biological processes. These metals can induce oxidative stress, disrupt cell membrane potential, and disturb cellular homeostasis. They also compete with essential proteins for binding sites, leading to dysfunction in vital processes [43,44]. Exposure to high levels of metals like arsenic (As), cadmium (Cd), and lead (Pb) has been strongly associated with adverse pregnancy outcomes, such as an increased risk of low birth weight, which can negatively impact long-term health [45,46,47].

In addition to these concerns, certain metals—particularly As, Cd, mercury (Hg), and Pb—are recognized neurotoxicants. Prenatal exposure to these metals has been linked to significant impairments in a child’s neurodevelopment and cognitive abilities, including delayed learning and memory deficits, which can have lifelong consequences [48,49,50,51]. Although the placenta provides some protection by partially blocking the transfer of Cd, it is far less effective at shielding the fetus from As, Pb, and Hg. These metals are able to cross the placental barrier with relative ease, posing serious risks to the developing fetus [52,53].

Cord blood DNA methylation (DNAm) at 5-methylcytosine sites reflects the newborn’s epigenetic state of fetal blood during pregnancy and is influenced by maternal exposures and genetics [54]. Cord blood DNAm has been associated with various maternal factors, including pre-pregnancy weight, weight gain during pregnancy, maternal smoking, depression, and prenatal arsenic exposure [55,56,57,58,59]. Additionally, previous studies have linked cord blood DNAm with a child’s weight at different life stages, body size, adiposity, and blood pressure percentiles [60,61,62].

Epigenetic analyses can be performed on newborn dried blood spots (NDBSs), which are collected in most hospitals for screening metabolic diseases. NDBSs are drops of whole blood collected on filter paper via a simple heel stick within 24–48 h after birth. Analyzing metals in NDBSs is a non-invasive method commonly used to assess perinatal exposure to toxic metals [63]. Similarly, DNA can be isolated from NDBSs to study DNA methylation changes resulting from perinatal exposure to toxic metals.

The following sections discuss how specific metals affect DNAm in umbilical cord blood. The subsections are as follows: Section 4.1. Arsenic exposure; Section 4.2. Mercury exposure; Section 4.3. Mercury and arsenic co-exposure; Section 4.4. Cadmium exposure; Section 4.5. Lead exposure; and Section 4.6. Exposure to mixtures of heavy metals.

We performed a systematic PubMed literature search using the keywords “Cord blood metals epigenetics”. We completed the search using the PubMed database (https://www.ncbi.nlm.nih.gov/pubmed (accessed on 31 August 2024)) and Google Scholar up to the end of August 2024. For individual heavy metals, a PubMed search was conducted separately with the keywords “prenatal arsenic exposure cord blood epigenetic changes”, “prenatal mercury exposure cord blood epigenetic changes”, “prenatal cadmium exposure cord blood epigenetic changes”, etc. For the health effects, PubMed search terms such as “long term health outcome of fetal exposure to heavy metals“ and "health outcome of fetal exposure to heavy metals dna methylation" were used. The selection of articles was made in compliance with the Preferred Reporting Items for Systematic reviews and Meta-Analyses (PRISMA) guidelines [64]. We have reported the null findings in Table 1.

### 4.1. Arsenic Exposure

The World Health Organization (WHO) has classified arsenic (As) and mercury (Hg) among the most dangerous substances, highlighting their serious threat to global public health [71]. Both As and Hg can cross the placenta, and numerous studies have demonstrated their potential neurotoxic effects on the developing fetus [72,73]. Human exposure to As, particularly through contaminated drinking water, affects over 2 million people worldwide [74]. Arsenic functions as an endocrine disruptor and a potent carcinogen. Several studies on humans and experimental animals suggest that in utero exposure to As can impair fetal and childhood health and development, potentially leading to cardiovascular diseases, cancer, and increased mortality during childhood or adulthood [75,76,77,78,79]. The epigenetic alterations caused by As exposure are often mediated through changes in DNA methylation (DNAm) patterns [80,81].

A study conducted in Bangladesh [82] with 127 mothers and their infants’ cord blood examined the DNAm of gene-related CpG sites in response to varying levels of maternal As exposure during both early and late pregnancy. This study was the first to investigate sex-specific genome-wide DNAm in cord blood concerning As exposure. The findings indicated that maternal As exposure in early gestation was associated with altered DNAm in newborns, even at low exposure levels (>50 µg/L) in urine. The associations were more pronounced in boys than in girls at the genome-wide DNAm level, with As exposure appearing to cause hypomethylation in boys. Of the top 500 differentially methylated CpG sites, 74 showed decreasing DNAm with increasing As exposure. The associations with arsenic (As) exposure were weaker during late gestation compared to early pregnancy. Pathway analysis indicated that cancer-related genes were disproportionately affected in boys, but this pattern was not observed in girls. In general, early prenatal exposure to As was linked to a reduction in DNA methylation (DNAm), specifically in boys.

### 4.2. Mercury Exposure

In 2019, researchers in Japan conducted a study to explore the relationship between prenatal exposure to total mercury (Hg) and DNA methylation (DNAm) in offspring, specifically investigating whether there were any sex-specific effects in male and female children [83]. The birth cohort study included 67 Japanese infants (27 males and 40 females), analyzing DNAm status in cord tissue and Hg concentrations in cord serum. The results showed that serum Hg concentrations at birth were associated with the DNAm of a single CpG site in the cord tissue of male infants only. This CpG site was linked to the HDHD1 gene on chromosome X, which encodes pseudouridine-5′-phosphatase (PUDP), a protein expressed in various organs, including the placenta and colon, as well as in lymphocytes [84]. No correlation was found between Hg concentration and DNAm status in cord tissue for female newborns. This study was groundbreaking in examining the potential relationship between prenatal mercury (Hg) exposure in cord serum and DNA methylation (DNAm) status in cord tissue, particularly highlighting sex-specific effects. The findings revealed a significant correlation between Hg concentrations in the cord serum of male offspring and the levels of DNAm within the HDHD1 gene.

Mercury is released into the environment from various natural and anthropogenic sources, such as coal-burning power plants. Since the Industrial Revolution, the concentration of mercury in the oceans has surged, tripling in amount [85]. In aquatic systems, anaerobic microbes biotransform Hg into methylmercury (MeHg), which bioaccumulates in species at higher trophic levels. Humans are primarily exposed to MeHg through fish consumption. MeHg can cross the placenta, making maternal consumption of large predatory fish, such as shark, tilefish, and king mackerel, a significant source of fetal exposure to Hg in utero [86,87]. Cord blood Hg levels are often higher than maternal blood levels, possibly due to higher levels of hemoglobin in cord blood, which binds free MeHg and increases active transport to the fetus [88,89,90,91] or decreased clearance due to low levels of glutathione (GSH), which binds MeHg for removal [91]. Increasing evidence links prenatal Hg exposure to poor cognitive development and behavioral disorders in children [92,93,94,95]. Even low Hg exposure levels have been associated with subclinical signs of autoimmunity among women of reproductive age in the USA [96]. Prenatal exposure to Hg, as measured in cord blood or toenails, has also been linked to DNAm changes in certain genes in cord blood cells [96,97].

A study investigated the connection between prenatal exposure to mercury (Hg) and DNA hydroxymethylation (DNAhm), an epigenetic modification essential for tissue differentiation and the development of the embryo [68]. DNAhm is produced by TET-family enzymes that convert 5-methylcytosine (DNAm) to 5-hydroxymethylcytosine (DNAhm). Besides converting DNAm to DNAhm, these enzymes also participate in erasing DNAm and are found in stem cells and some neurons. Research into the role of DNAhm in the epigenetics of environmental exposures represents an exciting new direction in epigenetic studies.

In the Hg study [68], the authors conducted an investigation involving a U.S. prebirth cohort to explore the associations between maternal red blood cell mercury (RBC-Hg) concentrations during the second trimester and global levels of 5-hydroxymethylcytosine (%-5hmC) and 5-methylcytosine (%-5mC) DNA content in blood samples collected at different life stages: birth (n = 306), early childhood (n = 68; ages 2.9 to 4.9 years), and mid-childhood (n = 260; ages 6.7 to 10.5 years). Their findings revealed that increased prenatal exposure to mercury in mothers, as measured in the second trimester, was associated with a reduction in global %-5hmC DNA content in cord blood, even after accounting for potential confounding variables. This association was maintained into early childhood but was not detected in the later years [68].

Another study explored the association between prenatal exposure to methylmercury (MeHg) and DNA methylation (DNAm) in cord blood by analyzing data from seven independent studies, which included a total of 1462 participants. Furthermore, the research also evaluated children aged 7 to 8 years, comprising a sample of 794 individuals [98]. The study found limited evidence of differential DNAm at the cg24184221 site in the MED31 gene in relation to prenatal MeHg exposure. Additionally, a limited association was observed at the cg15288800 site in MED31 in child blood. The findings suggest that DNAm levels in MED31 may be disrupted in response to in utero MeHg exposure, with effects persisting into early childhood [98].

Moreover, the study identified associations between prenatal exposure to methylmercury (MeHg) and DNA methylation (DNAm) levels in children’s blood at two specific CpG sites. The first, cg12204245, is located in the GRK1 gene, which encodes the rhodopsin kinase protein that is predominantly found in the retina and is essential for proper retinal function. The second site, cg02212000, is found in the GGH gene, which is widely expressed in the body and plays a key role in glutathione metabolism and pathways related to the innate immune system [98]. These findings enhance the understanding of the potential mechanisms underlying the adverse health effects of MeHg exposure by highlighting the role of DNAm in several genes involved in growth and cell cycle processes during fetal development.

A study sought to determine if prenatal exposure to mercury (Hg), a well-known neurotoxic metal, is connected to reduced cognitive performance in children [99]. The study investigated epigenome-wide DNAm differences associated with maternal prenatal blood Hg levels in 321 cord blood DNA samples, examining the persistence of these alterations during early (n = 75; 2.9–4.9 years) and mid-childhood (n = 291; 6.7–10.5 years). DNAm changes in a genomic region of the Paraoxonase 1 (PON1) gene were associated with prenatal Hg exposure in umbilical cord blood from males but not females. This association persisted into early and mid-childhood, with cord blood DNAm at the PON1 locus predicting lower cognitive test scores measured during early childhood [99]. 

In summary, these results imply that moderate exposure to mercury (Hg) during pregnancy can induce sex-specific functional epigenetic modifications that persist throughout childhood. These alterations are not merely biological markers; they are associated with variations in cognitive performance in children. This suggests that the impact of prenatal mercury exposure may have lasting effects on brain development and function, potentially leading to differences in cognitive abilities based on sex. Such findings underscore the importance of monitoring and minimizing mercury exposure during pregnancy to safeguard the neurodevelopment of offspring.

### 4.3. Mercury and Arsenic Co-Exposure

A study investigated the effects of prenatal exposure to mercury (Hg) and its interaction with arsenic (As) on DNA methylation (DNAm) in cord blood. The study included 138 mother–infant pairs, primarily white (98%) [96]. Postpartum maternal toenail mercury levels, prenatal urinary arsenic concentrations, and newborn cord blood samples were analyzed using the Illumina Infinium Methylation 450 (HM450K) array to investigate the effects of these toxic exposures on DNA methylation patterns. The study also estimated the composition of white blood cells based on the DNA methylation measurements obtained from the cord blood samples. The findings revealed a significant association between increased toenail mercury concentration and a 2.5% decrease (95% CI: 5.0%, 1.0%) in the estimated proportion of monocytes in the newborns. This decrease suggests that higher mercury exposure may affect the immune system’s development by altering white blood cell composition. In an interesting sex-specific observation, a 3.5% increase in B-cell levels was noted exclusively in female infants, indicating potential differences in how male and female infants respond to mercury exposure.

Among the top 100 CpGs associated with toenail mercury levels, there was a significant enrichment of loci located in the north shore regions of CpG islands, with a statistical significance of *p* = 0.049. Notably, 85% of these loci were hypermethylated, suggesting that mercury exposure may lead to epigenetic modifications that could have long-term implications for health. Furthermore, when examining the top 100 CpGs related to the interaction between arsenic (As) and mercury (Hg), a substantial proportion of the loci were found within CpG islands (*p* = 0.045) and south shore regions (*p* = 0.009), with all of these loci also exhibiting hypermethylation. This highlights the complex interplay between these two toxic metals and their potential combined effects on DNA methylation and, consequently, on health outcomes in children [96]. This study provides evidence that co-exposure to Hg and As in utero may influence the epigenome of cord blood, altering the underlying leukocyte composition by decreasing monocyte proportions and increasing B-cell populations in female infants.

In low- and middle-income countries, occupational exposure to arsenic (As) and mercury (Hg) in artisanal and small-scale gold mining (ASGM) communities poses a significant public health challenge. A study was conducted to examine the association between prenatal maternal exposure to As and Hg and various birth outcomes in both ASGM and non-ASGM communities in Northern Tanzania. This prospective longitudinal study recruited a total of 961 pregnant women, with 788 participants from ASGM areas and 173 from non-ASGM areas. During the second trimester of pregnancy, maternal spot urine samples and dried blood spots were collected to measure levels of total arsenic (T-As) and total mercury (T-Hg). The researchers focused on gathering data related to adverse birth outcomes, categorizing these into five key areas: spontaneous abortion, stillbirth, preterm birth, low birth weight, and visible congenital anomalies. The findings revealed a concerning disparity between the two groups, with the risk of adverse birth outcomes significantly higher in ASGM areas (54%) compared to non-ASGM areas (27%). Specifically, increased exposure to arsenic and mercury during pregnancy was associated with a greater incidence of several adverse birth outcomes, including spontaneous abortion, stillbirth, preterm birth, low birth weight, and visible congenital anomalies among the Tanzanian women studied. These results underscore the critical need for public health interventions and policies aimed at reducing exposure to these toxic metals in ASGM communities to protect maternal and child health [100].

### 4.4. Cadmium Exposure

Cadmium (Cd) is a heavy metal that is classified among the top ten chemicals on the Agency for Toxic Substances and Disease Registry (ATSDR) priority list of hazardous substances. This classification underscores the significant concern surrounding cadmium due to its widespread environmental presence and its potential to pose serious health risks to humans. Cadmium is primarily found in industrial settings, particularly in the production of batteries, pigments, coatings, and plastics. It can also be released into the environment through mining, smelting, and the burning of fossil fuels. Due to its persistent nature, cadmium can accumulate in the soil and water, leading to contamination of food sources, particularly crops. Exposure to cadmium can occur through inhalation, ingestion, or skin contact, and it is particularly concerning for vulnerable populations, including pregnant women and children. The health effects of cadmium exposure can be severe, including kidney damage, bone fragility, and increased risk of cancer. Additionally, chronic exposure has been linked to various adverse effects on respiratory and cardiovascular health. Given these potential dangers, cadmium’s ranking on the ATSDR priority list highlights the urgent need for continued monitoring, regulation, and public awareness efforts to mitigate exposure and protect public health [101].

Cadmium is widely present in the environment, including industrial byproducts, contaminated water or soil, and tobacco products [102]. Cd is recognized as a carcinogen in various tissues such as the lung, liver, prostate, kidney, bladder, stomach, and pancreas [103,104]. Prenatal exposure to Cd has been linked to developmental issues, negatively affecting fetal growth parameters like birth weight, length, head circumference, and height, as well as leading to cognitive developmental defects later in life [46,52,93,105,106,107,108,109,110]. Pregnant women can be exposed to cadmium (Cd) through various routes, including smoking, air pollution, and contaminated food and water. Smoking is a significant source, as tobacco plants can absorb cadmium from the soil, leading to higher levels in cigarette smoke. Air pollution, especially in urban areas, contributes to exposure through industrial emissions and vehicle exhaust [111]. Several reports indicate that prenatal Cd exposure affects gene expression/function via DNA methylation (DNAm) [112,113], with Cd-associated DNAm observed in specific CpG sites within genes related to organ development, glucocorticoid synthesis, and cell death [114].

One study analyzed more than 16,000 promoter-based CpG islands in the DNA of both mothers and their newborns. The researchers found significant alterations in DNAm levels linked to cadmium exposure, identifying 61 genes in fetal DNA and 92 genes in maternal DNA that exhibited changes in their methylation patterns. Notably, the majority of these differentially methylated genes demonstrated increased levels of methylation, indicating hypermethylation due to cadmium exposure. Furthermore, the study delved into the underlying mechanisms of these methylation changes by identifying conserved DNA motifs with sequence similarities to specific transcription factor binding sites within the altered CpG islands. This finding suggests the presence of distinct DNAm patterns or “footprints” in both fetal and maternal DNA that are specifically associated with cadmium exposure. These unique patterns may have implications for understanding how environmental toxins like cadmium can influence gene expression and contribute to adverse health outcomes in both mothers and their children [115].

Another study conducted in Korea investigated the relationships between trimester-specific prenatal cadmium (Cd) exposure and DNA methylation (DNAm) in cord blood samples from infants. This research utilized an epigenome-wide association study (EWAS) to examine how variations in cadmium concentrations during different stages of pregnancy influenced methylation patterns in newborns. To assess cadmium exposure, researchers measured Cd concentrations in maternal blood samples collected at various points throughout pregnancy, as well as in the cord blood of infants. The measurement technique employed was atomic absorption spectrometry, a sensitive method capable of accurately quantifying trace metal levels. Additionally, the analysis of DNAm was performed using the Illumina (San Diego, CA, USA) HM450K arrays, which allow for comprehensive profiling of methylation across numerous CpG sites. The findings of the study revealed that exposure to cadmium during early pregnancy was significantly associated with two specific differentially methylated CpG sites: cg05537752, which is annotated to the ATP9A gene, and cg24904393, associated with an unspecified gene. These results underscore the importance of the timing of cadmium exposure, as the study indicates that the fetus’s susceptibility to cadmium-induced epigenetic changes varies across different trimesters of pregnancy [116].

A study conducted in rural Bangladesh investigated the relationships between prenatal cadmium (Cd) exposure, DNA methylation (DNAm), and birth weight in a cohort of 127 mother–child pairs. This research aimed to understand how maternal exposure to cadmium during pregnancy could influence the epigenetic landscape of newborns and potentially impact their health outcomes, particularly regarding birth weight. Among the participants, 56 children were evaluated when they reached 4.5 years of age. The findings revealed a significant association between maternal cadmium exposure and sex-specific DNAm changes in the cord blood of infants. Notably, 96% of the top 500 CpG sites identified in the analysis exhibited positive correlations with cadmium exposure in boys. In contrast, most of the associations observed in girls were inverse, indicating a complex interaction between sex and environmental exposure. Further analysis indicated that the DNAm changes observed in girls were over-represented in genes related to organ development, morphology, and bone mineralization. This suggests that prenatal cadmium exposure may have specific impacts on the developmental processes crucial for proper growth and formation in female infants. Conversely, the changes in boys were primarily linked to genes associated with cell death, which may imply different biological responses to cadmium exposure between the sexes. Moreover, the study found that several CpG sites that were positively associated with cadmium exposure were inversely correlated with birth weight. This relationship indicates that early-life exposure to cadmium could lead to alterations in DNA methylation that negatively affect birth weight. The findings suggest that these epigenetic changes may play a role in the mechanisms by which cadmium exposure impacts fetal development and health outcomes [117].

A follow-up study in rural Bangladesh assessed whether gestational Cd-related DNAm changes persist from birth to nine years of age [118]. In this longitudinal cohort study, 127 mother–child dyads were examined, with measurements of Cd concentrations in maternal blood and child urine at nine years. The study identified ten differentially methylated regions (DMRs) in cord blood linked to gestational Cd exposure, with one DMR validated in an independent sample of nine-year-olds. The study found that regional DNAm changes associated with gestational Cd exposure persisted up to prepubertal age, suggesting a lasting epigenetic impact of prenatal Cd exposure [118].

Another study focused on Cd exposure’s impact on DNAm in newborn cord blood and maternal blood, using bisulfite sequencing of samples selected based on maternal blood Cd levels during the first trimester [119]. The study identified 641 Cd-associated DMRs in newborn cord blood and 1945 in maternal blood. These DMRs were more common at maternally methylated imprinting control regions (ICRs) than at similar non-imprinted loci, indicating a higher sensitivity of ICRs to Cd exposure. The findings suggest that ICRs may be particularly susceptible to Cd, warranting further investigation into these regulatory sequences’ roles in response to environmental stressors [119].

A Korean study explored the association between maternal and prenatal heavy metal exposure, including Cd and gestational age, an indicator of preterm birth risk (n = 1751 pregnant women from 2006 to 2010) [120]. DNAm analysis using cord blood DNA from 384 participants revealed that maternal Cd exposure was associated with a reduction in gestational age, mediated through altered DNAm at a specific CpG site, cg21010642, related to a gene involved in early embryonic development. These findings suggest that irregular DNAm patterns at this site may contribute to preterm birth by disrupting normal biological processes [120].

### 4.5. Lead Exposure

Lead (Pb) is a pervasive and highly toxic heavy metal that can easily cross the placenta, posing a significant risk to the developing fetus [121,122]. Even at low levels, Pb exposure can lead to considerable cognitive and neurobehavioral deficits in children [123]. It has also been linked to spontaneous abortions [124], preterm birth [125,126], and low birth weight [127,128], all of which can result in adverse health outcomes throughout life [129,130].

A 2015 study conducted by our group revealed that a child’s neonatal blood DNA methylation (DNAm) profile could be significantly affected by the mother’s neonatal blood Pb levels (BLL) [131]. In this study, 35 mother–infant pairs from Detroit, MI, USA, were recruited, and whole blood Pb and DNAm levels at over 450,000 loci were analyzed in both current blood and neonatal blood from the mother and child. The study examined three generations: F1 represented the maternal grandmother, F2 represented the mother as a fetus, and F3 the child born to the mature mother. The study found that mothers with high neonatal BLL had children with altered DNAm at 564 loci in their neonatal blood, suggesting that exposure to environmental toxins like Pb can cause epigenetic changes in the newborn blood of a grandchild of an exposed pregnant woman [131].

In a subsequent study conducted by our research group, we delved deeper into the relationship between lead (Pb) exposure and the epigenetic modifications known as 5-methylcytosine (5mC) and 5-hydroxymethylcytosine (5hmC) during early development. Utilizing an innovative approach, we employed a modified version of the HM450K array, referred to as the HMeDIP-450K assay, to analyze changes in the 5hmC profile in an in vitro model of human embryonic stem cells exposed to lead. Our findings revealed clusters of correlated, adjacent CpG sites that responded to Pb exposure, indicating a specific pattern of epigenetic modification linked to the presence of this toxic metal. This clustering suggests that certain genomic regions are particularly sensitive to lead, potentially highlighting critical areas where Pb exposure could have significant developmental implications [132].

In addition to the in vitro studies, we expanded our investigation to examine Pb-dependent changes in high-density 5hmC regions within umbilical cord blood DNA. This analysis was conducted using samples from 48 mother–infant pairs drawn from the Early Life Exposure in Mexico to Environmental Toxicants (ELEMENT) cohort. Among these pairs, we included an equal representation of 24 male and 24 female children, selected randomly from the first and fourth quartiles of Pb levels to ensure a balanced examination of exposure effects across genders. Our research demonstrated that changes in 5hmC and 5mC profiles associated with lead exposure could be classified into two categories: sex-dependent and sex-independent modifications. Interestingly, the differential sites exhibiting 5mC changes emerged as more reliable markers for identifying sex-dependent alterations in response to lead exposure. This distinction is crucial as it suggests that boys and girls may react differently to environmental toxins, further emphasizing the need to consider sex as a biological variable in toxicological research. Through this study, we identified several genomic loci for both 5hmC and 5mC modifications as potential early biomarkers of prenatal Pb exposure. These biomarkers could serve as important tools for monitoring and assessing the impact of environmental toxins on fetal development, ultimately aiding in public health efforts to mitigate the effects of lead exposure during critical developmental windows. By understanding the epigenetic consequences of lead exposure, we can better inform strategies aimed at protecting vulnerable populations, particularly pregnant women and infants, from the harmful effects of this toxic metal [132].

A study by Pilsner et al. (2009) aimed to determine whether prenatal Pb exposure is associated with alterations in genomic DNAm of leukocyte DNA from umbilical cord blood (UCB) samples [66]. The researchers measured genomic DNAm, specifically Alu and LINE-1 (long interspersed nuclear element-1) methylation, via pyrosequencing in 103 UCB samples from the ELEMENT study group’s biorepository. Prenatal Pb exposure was assessed by measuring maternal bone lead levels at the mid-tibial shaft and patella using a 109Cd K-shell X-ray fluorescence instrument. The study found an inverse dose–response relationship: higher patella Pb levels were associated with decreased LINE-1 DNAm in cord blood, and higher tibia Pb levels were associated with decreased Alu DNAm (*p* = 0.05). In mixed effects regression models, maternal tibia Pb was negatively correlated with umbilical cord Alu DNAm (*p* = 0.01) [66].

In conclusion, prenatal Pb exposure is inversely associated with genomic DNAm in cord blood. These findings suggest that the epigenome of the developing fetus can be influenced by maternal cumulative Pb burdens, potentially affecting long-term epigenetic programming and disease susceptibility across the lifespan.

### 4.6. Exposure to Mixtures of Heavy Metals

A prospective birth cohort study conducted in Japan explored the effects of neurotoxicants on child development, specifically focusing on the relationships between prenatal exposure to various toxic metals and global DNA methylation (DNAm) in umbilical cord blood (UCB) DNA. This comprehensive study evaluated the potential associations of five specific heavy metals—arsenic (As), cadmium (Cd), mercury (Hg), lead (Pb), and antimony (Sb)—as well as polychlorinated biphenyls (PCBs) within a cohort of 166 participants. The primary aim of the research was to investigate how these toxic substances might influence the epigenetic landscape of newborns, particularly through the analysis of DNAm markers, namely 5-methylcytosine (5mC) and 5-hydroxymethylcytosine (5hmC). By examining these markers in umbilical cord blood samples, the study sought to provide insights into how prenatal exposure to environmental toxins could impact developmental processes at the molecular level. The findings of the study revealed that the content of 5mC in cord blood DNA was positively correlated with the levels of lead (Pb) and antimony (Sb), while no significant correlation was found with polychlorinated biphenyls (PCBs). This suggests that Pb and Sb exposure may directly influence the methylation patterns in the DNA of infants at birth, indicating potential pathways through which these metals could affect child development. Additionally, the study observed significant positive correlations between maternal age, Pb levels, and the content of 5hmC. This relationship highlights the complexity of factors that may influence epigenetic modifications, suggesting that older maternal age might interact with toxic metal exposure to further affect the epigenetic profile of the newborn. Further analysis using multiple regression techniques demonstrated consistent positive associations between Pb and Sb levels and the content of both 5mC and 5hmC. These results underscore the potential of global DNA methylation as a promising biomarker for assessing prenatal exposure to these neurotoxic metals. By identifying specific epigenetic changes associated with toxic metal exposure, this research contributes to a growing body of evidence linking environmental contaminants to developmental outcomes [133].

In a significant study conducted in China, researchers investigated the impact of maternal exposure to heavy metals derived from electronic waste (e-waste) on neonatal DNA methylation (DNAm) patterns. This research is crucial as e-waste recycling areas often have heightened levels of toxic metals, which can pose substantial health risks to vulnerable populations, particularly pregnant women and their newborns. The study was carried out in two distinct geographical locations: Guiyu, a well-known e-waste recycling hub, and Haojiang, a non-e-waste area that served as a control site. A total of 204 participants were enrolled in the study, comprising 101 mothers from Guiyu and 103 from Haojiang. To evaluate the extent of heavy metal exposure, umbilical cord blood (UCB) samples were collected from each participant, and concentrations of lead (Pb), cadmium (Cd), manganese (Mn), and chromium (Cr) were meticulously measured. Using the Illumina HM450K array, the researchers conducted an epigenome-wide analysis of DNAm across 473,844 CpG sites in the collected UCB samples. Through advanced bioinformatics techniques, they identified 125 differentially methylated CpG sites corresponding to 79 genes. Notably, the analysis revealed significantly higher concentrations of heavy metals in UCB from the mothers exposed to e-waste, particularly highlighting the correlation between elevated Pb levels and changes in DNAm patterns. The genes identified as differentially methylated were associated with critical biological processes, including calcium ion binding, cell adhesion, and embryonic morphogenesis. Additionally, these genes were linked to essential signaling pathways, such as NF-κB activation, adherens junctions, TGF-beta signaling, and apoptosis. Among the differentially methylated genes, BAI1 and CTNNA2 were of particular interest. BAI1, which is implicated in neuronal differentiation and development, was found to be hypermethylated in relation to maternal Pb exposure. Conversely, CTNNA2, another gene associated with neurodevelopment, exhibited hypomethylation under similar exposure conditions. The findings of this study strongly suggest that maternal exposure to heavy metals from e-waste, especially lead, during pregnancy can lead to significant alterations in DNAm in newborns. These changes in DNAm are likely to affect genes integral to neurodevelopment, raising concerns about the long-term implications for cognitive and behavioral outcomes in children [134].

In a comprehensive study conducted in China, researchers investigated the associations between individual metal exposures, mixtures of metals, oxidative stress, and DNA methylation (DNAm) among a cohort of 113 pregnant women. This research is particularly pertinent as it seeks to understand how environmental exposures to heavy metals can affect maternal health and potentially impact fetal development. The study focused on a diverse panel of 16 metals, which included arsenic (As), cadmium (Cd), thallium (Tl), barium (Ba), nickel (Ni), vanadium (V), cobalt (Co), zinc (Zn), copper (Cu), selenium (Se), and molybdenum (Mo). Alongside measuring these metals, the researchers also evaluated three biomarkers of oxidative stress in the urine of the participants: 8-hydroxydeoxyguanosine (8-OHdG), 4-hydroxy-2-nonenal-mercapturic acid (HNE-MA), and 8-isoprostaglandin F2α (8-isoPGF2α). These biomarkers are critical indicators of oxidative damage and provide insight into the body’s response to environmental stressors. In addition to assessing metal concentrations, the researchers analyzed global DNAm markers in the cord blood of newborns, specifically targeting Alu and LINE-1 sequences, which are commonly used to evaluate overall methylation patterns. The single-metal analyses revealed positive associations between 11 of the metals studied and at least one oxidative stress biomarker, indicating that exposure to these metals may elevate oxidative stress levels during pregnancy. This finding highlights the potential risks associated with even low-level exposures to heavy metals. Further exploration through mixture analyses demonstrated that combinations of metals were positively correlated with oxidative stress markers, particularly with 8-OHdG and 8-isoPGF2α. Among the metals analyzed, selenium (Se) emerged as the most significant factor influencing oxidative stress, suggesting that its presence in metal mixtures could be particularly impactful [135].

A study in the USA explored the relationships between various metals—specifically arsenic (As), barium (Ba), cadmium (Cd), chromium (Cr), cesium (Cs), copper (Cu), mercury (Hg), magnesium (Mg), manganese (Mn), lead (Pb), selenium (Se), and zinc (Zn)—and their impact on DNA methylation in cord blood, as well as the persistence of these effects into mid-childhood. The study analyzed first-trimester maternal red blood cell (RBC) samples to assess metal concentrations, correlating these levels with differentially methylated positions (DMPs) and regions (DMRs) in the offspring. In this investigation, metal concentrations were meticulously measured in the RBCs of pregnant women during the first trimester. DNA methylation analysis was then performed on both cord blood (with a sample size of 361 participants) and blood samples from mid-childhood (with 333 participants aged between 6 and 10 years) utilizing the HM450K array technology. The results revealed notable findings, including the identification of two specific CpG sites in cord blood where DNA methylation was significantly associated with metal exposure during the first trimester. The first CpG site, cg02042823, was linked to manganese (Mn), while the second site, cg20608990, was associated with lead (Pb). Importantly, these associations persisted into mid-childhood, suggesting a long-lasting impact of early metal exposure on DNA methylation patterns. The study also noted sex-specific associations regarding the influence of Mn exposure on cord blood DNA methylation. In females, nine CpGs were affected, whereas only two CpGs showed changes in males. Some of these sex-specific associations continued to persist into mid-childhood, highlighting the nuanced ways in which prenatal metal exposure can differentially affect males and females [136].

A study conducted in Mexico City delved into the effects of prenatal exposure to various metals—specifically arsenic (As), copper (Cu), mercury (Hg), manganese (Mn), molybdenum (Mo), lead (Pb), selenium (Se), and zinc (Zn)—on both global and gene-specific DNA methylation (DNAm) as well as mRNA expression in a cohort of 181 umbilical cord blood (UCB) samples. This research aimed to illuminate the intricate relationship between environmental metal exposure during pregnancy and its subsequent impact on epigenetic and gene expression profiles in newborns. The study meticulously assessed global DNA methylation using LINE-1 sequences, a repetitive element often utilized as a marker for overall genomic methylation status. Additionally, it focused on promoter methylation of specific genes involved in DNA repair and antioxidant responses, namely OGG1, PARP1, and Nrf2, utilizing a technique called pyrosequencing. This method allowed for precise quantification of DNAm levels at particular sites within these genes. Through multiple regression analyses, the researchers found significant associations between the DNAm of LINE-1, Nrf2, OGG1, and PARP1 and the levels of both potentially toxic metals (such as As, Hg, Mn, Mo, and Pb) and essential trace elements (including Cu, Se, and Zn). These findings indicate that exposure to a mixture of metals can have complex interactions affecting DNAm patterns in developing fetuses. Notably, the study observed that increased DNAm at specific sites within the OGG1 gene was linked to decreased mRNA expression levels, suggesting that higher methylation could be inhibiting the gene’s transcription. Similarly, several sites within the *PARP1* gene displayed reduced mRNA expression associated with increased DNAm. However, an intriguing finding was that DNAm at the *PARP1* CpG7 site exhibited a positive correlation with mRNA levels, indicating that the relationship between DNAm and gene expression may not always be straightforward and can vary depending on the specific genomic context. In terms of the *Nrf2* gene, no significant associations were found between its expression levels and the DNAm status at the analyzed CpG sites. This lack of correlation suggests that the regulation of *Nrf2* may involve additional layers of complexity or alternative regulatory mechanisms that are not solely dependent on DNAm modifications [137]. Overall, the studies in this section concluded that DNAm can be influenced by prenatal metal exposure, potentially altering gene expression and reducing the capacity for DNA damage repair in newborns.

## 5. Short-Term and Long-Term Health Outcomes of Fetal Exposures to Heavy Metals in Relation DNA Methylation

Growing fetuses and young children are significantly more sensitive to mercury (Hg) exposure than adults due to their immature detoxification systems and developing immune responses. Fetuses rely on maternal metabolism for nutrient delivery and detoxification, making them particularly vulnerable to accumulating toxic mercury levels. The placenta provides some protection, but it is not entirely effective in blocking mercury transfer, resulting in potential neurological impairments and developmental issues. In infants and young children, key organs involved in detoxification, such as the liver and kidneys, are still maturing. This developmental immaturity hinders their ability to metabolize and excrete mercury efficiently. Additionally, children’s higher metabolic rates and relative body surface area increase their overall exposure to toxins [138].

In 2006, a longitudinal cohort study was conducted in Korea, aimed at investigating the relationship between mercury (Hg) exposure during prenatal and early childhood periods and the development of autistic behaviors in preschool-age children. This study involved 458 mother–child pairs, providing a robust sample size to analyze the long-term effects of Hg exposure on neurodevelopment. The researchers measured blood Hg levels at multiple time points: during early pregnancy, late pregnancy, in cord blood at birth, and subsequently when the children reached the ages of 2 and 3 years. By collecting data at these critical developmental stages, the study aimed to establish a clear timeline of Hg exposure and its potential impact on the children’s behavioral outcomes. The focus on both prenatal and early childhood Hg exposure was particularly relevant, as these periods are known to be crucial for brain development. The study sought to determine whether higher levels of Hg in maternal blood during pregnancy or in the child’s blood during early childhood were associated with an increased risk of developing autistic behaviors. Through statistical analysis, the study assessed the prevalence of autistic traits in the children and explored potential correlations with the measured Hg levels. The findings from this research could have significant implications for public health policies regarding environmental exposures during pregnancy and early childhood, particularly in populations at risk of mercury contamination [139]. Autistic behaviors were assessed at age 5 using the Social Responsiveness Scale (SRS). The study included more boys (55.2%) than girls. The results showed that Hg concentrations at late pregnancy, in cord blood, and at ages 2 and 3 were linked to autistic behaviors at age 5. A sex-related difference in response was observed, suggesting that biological pathways involving endocrine disruption could be associated with autistic behavior due to Hg exposure [140]. A previously discussed study in this review indicated that prenatal Hg exposure disrupted neurodevelopmental processes through fetal epigenetic programming, leading to lower cognitive performance during childhood [99]. These findings underscore the need for public attention to environmental Hg exposure during pregnancy and early childhood.

Maternal fish consumption has complex implications for fetal development, as it exposes developing fetuses to mercury (Hg) while simultaneously providing a vital source of long-chain polyunsaturated fatty acids (LCPUFAs), which are critical for normal fetal growth and neurodevelopment. A study was conducted to investigate the combined effects of LCPUFA intake and Hg exposure from fish consumption during pregnancy on newborn brain development and subsequent child neurodevelopment. This research involved 257 mother–infant pairs from the northern Adriatic coastal region, an area where fish is a common part of the diet. The study meticulously measured concentrations of total mercury (THg) and key LCPUFAs, specifically docosahexaenoic acid (DHA) and arachidonic acid (ARA), in umbilical cord blood (UCB). By analyzing these biomarkers, the researchers aimed to gain insight into how maternal dietary habits influenced fetal neurodevelopment. To assess the impacts of Hg exposure and LCPUFA levels, neonatal cranial sonography was performed on newborns at just three days of age, allowing for early morphological assessments of brain structure. Additionally, cognitive, motor, and language skills were evaluated at 18 months using the Bayley Scales of Infant and Toddler Development, Third Edition. Participants were categorized into two distinct groups based on the concentration of THg in their UCB. Those exposed to higher levels of Hg (greater than 5.8 μg/L) were compared against a control group with lower exposure levels (less than 5.8 μg/L). The findings revealed significant morphological changes in the brains of newborns exposed to Hg, particularly notable differences in cerebellum length and the width of the gyrus frontalis superior between the two groups. Specifically, the exposed group exhibited a wider frontal gyrus but a narrower cerebellum, highlighting the potential structural alterations associated with Hg exposure. Moreover, when considering the combined effects of LCPUFA intake and Hg exposure, the impact on brain morphology was somewhat mitigated, indicating that LCPUFAs might offer some protective benefits against the negative consequences of Hg. Interestingly, while maternal fish consumption correlated with increased levels of LCPUFAs, the study suggested that the consumption of fish itself—rather than LCPUFAs alone—was linked to enhanced language development at 18 months of age [141]. Given that seafood is the primary DHA source, further research into DHA supplementation’s potential to improve cognitive development in children is warranted.

Early-life exposure to lead (Pb) has been consistently linked to various neurodevelopmental deficits as well as toxicity within the hematopoietic system. A comprehensive study conducted with a prebirth cohort in the United States aimed to delve deeper into the long-term health effects associated with lead exposure, particularly focusing on potential sex-specific differences in outcomes. This study included a total of 268 mother–infant pairs, all of whom were characterized by relatively low levels of lead exposure. The researchers set out to examine how even low levels of Pb exposure could influence the health and development of children over time. Given that neurodevelopmental processes are critically sensitive during the early stages of life, the study sought to identify any lasting impacts that lead might have on cognitive and physical development as the children grew [142]. Pb concentrations in red blood cells (RBCs) from prenatal maternal blood samples were measured, and genome-wide DNAm levels at 482,397 CpG loci in UCB were analyzed using HM450K arrays. The study also extrapolated prenatal Pb exposure–cord blood DNAm associations using mid-childhood data for 240 children (7–10 years). Significant sex-specific differences were found, with RBC Pb levels affecting DNAm in more CpGs of female infants than males. In utero Pb exposure altered DNAm levels at CpGs located near crucial regulatory genes for hematopoietic and nervous functions, such as CLEC11A and DNHD1. CLEC11A encodes a secreted sulfated glycoprotein (CLEC11A) that acts as a growth factor for primitive hematopoietic progenitor cells [143] and enhances the proliferation and differentiation of hematopoietic precursor cells from several lineages [144]. DNHD1 (DYNC1H1), highly expressed in the brain, is a member of the dynein heavy chain family [142]. DNAm changes in the DNHD1 gene may result from in utero Pb exposure. Previous studies have linked Pb exposure to impaired cognitive and behavioral functions [92,145,146]. The significant inverse association between Pb exposure and DNAm changes was persistent through mid-childhood in female infants, highlighting the potential for long-term epigenetic programming effects.

The prevalence of high levels of lead (Pb) exposure in various countries, particularly in Mexico, has raised significant public health concerns and prompted extensive investigations into effective and affordable intervention strategies. Given the well-documented detrimental effects of lead exposure on health, especially in vulnerable populations such as children, there is an urgent need to develop and implement strategies that can mitigate these adverse effects at a population level [147]. Enhanced bone mobilization, particularly during critical developmental periods such as in utero and postpartum lactation, plays a significant role in the dynamics of lead (Pb) storage and exposure. This phenomenon is especially relevant in understanding how maternal and infant health can be affected by Pb [148,149]. The fetal and infancy stages are the most vulnerable to environmental exposures, affecting a wide range of health outcomes [129,150,151].

Maternal bones, particularly in regions such as the patella and tibia, can act as a significant endogenous source of lead (Pb) during critical life stages, including pregnancy and lactation. Over the years, lead can accumulate in the skeletal system due to various environmental exposures, including lead-based paints, contaminated water, and industrial pollutants. During pregnancy, the physiological demands placed on the mother’s body, particularly in supporting fetal development, trigger the mobilization of minerals, including calcium and, unfortunately, lead from bone stores into the bloodstream. To mitigate the risk of Pb mobilization from maternal bones and its adverse effects, daily supplementation with elemental calcium (Ca) has been shown to be effective. Studies indicate that a daily intake of 1200 mg of elemental calcium during pregnancy and lactation can significantly reduce the resorption of lead from maternal bone stores. This reduction occurs because adequate calcium levels help stabilize the bone mineral matrix and minimize the release of lead that can occur during periods of increased bone remodeling. By maintaining higher calcium levels, the body can prioritize calcium mobilization while limiting the release of lead, thereby decreasing circulating Pb levels in the mother’s bloodstream. Furthermore, enhancing maternal calcium intake not only serves to reduce Pb mobilization but also supports overall maternal health and fetal development. Adequate calcium is vital for bone health, muscle function, and proper nerve transmission. Ensuring that pregnant and lactating women receive adequate calcium can improve maternal health outcomes and promote better fetal growth and development.

Early-life exposure to lead (Pb) has been increasingly recognized as a significant risk factor for a range of adverse health outcomes in children. Numerous studies have established associations between early Pb exposure and neurocognitive deficits, leading to challenges in learning, memory, and overall cognitive function. Additionally, behavioral disorders such as attention deficit hyperactivity disorder (ADHD), anxiety, and aggression have been linked to elevated levels of Pb in early development. The timing of Pb exposure during critical developmental windows is another crucial factor to consider. Research indicates that exposure during specific periods, such as prenatal and early postnatal stages, may have more pronounced effects compared to exposure later in life. These critical windows of development are characterized by rapid growth and significant neurodevelopmental processes, making them particularly vulnerable to environmental toxins like Pb. Understanding these timing effects can help identify at-risk populations and inform prevention strategies. Given the significant implications of early Pb exposure, future research should continue to explore the long-term consequences of toxicant exposure during the perinatal period. It is essential to investigate how these early exposures may influence outcomes not only in childhood but also into adolescence and young adulthood. For instance, understanding the impact of early Pb exposure on academic performance, social behavior, and mental health during the teenage years can provide valuable insights into the long-term trajectory of affected individuals.

Early-life exposure to lead (Pb) has emerged as a significant concern, particularly regarding its potential impact on cardiac development and long-term heart health. Research indicates that Pb exposure during critical periods of growth can adversely affect various organ systems, including the heart, raising concerns about possible cardiac damage that may manifest in adulthood. However, the specific mechanisms through which early-life Pb exposure influences fetal heart development and the subsequent long-term cardiac outcomes remain poorly understood. To investigate these critical connections, a detailed study was conducted using pregnant ICR mice as a model to explore the effects of Pb exposure on heart development. In this experiment, lead acetate trihydrate was administered to the pregnant mice at a dose of 50 mg/kg per day via oral gavage, beginning on gestation day 1.5 and continuing until the offspring were weaned. This dosage was intentionally selected to reflect the low levels of Pb that have been previously associated with children suffering from congenital heart disease, which were found to be around 3.38 μg/L in earlier studies. By using a relevant animal model and dose, the researchers aimed to simulate conditions that may be encountered in human populations exposed to low levels of Pb, particularly in the context of developing fetuses. Furthermore, the study incorporated a second hit model to assess the interaction between early-life Pb exposure and subsequent stressors on cardiac health. In this model, the offspring, at 4 weeks of age, were divided into two groups. One group received sterile saline as a control treatment, while the other group was administered angiotensin II (Ang II) for 4 weeks until euthanasia. The use of Ang II, a peptide hormone that plays a crucial role in regulating blood pressure and fluid balance, allowed the researchers to simulate additional physiological stress on the cardiovascular system. By examining the effects of both Pb exposure and the second hit of Ang II on the developing hearts of the offspring, this study sought to elucidate the potential pathways through which early Pb exposure might predispose individuals to cardiovascular issues later in life. This approach is particularly important, as it may reveal whether Pb exposure during critical developmental windows can lead to structural and functional changes in the heart that contribute to long-term health risks, such as hypertension or heart disease [152].

A human study was conducted to explore the potential links between maternal cadmium (Cd) levels during early pregnancy and the DNA methylation (DNAm) patterns in offspring, specifically focusing on regulatory sequences of genomically imprinted genes. Genomic imprinting is a critical epigenetic phenomenon where certain genes are expressed in a parent-of-origin-specific manner, and alterations in their regulation can have significant implications for fetal development and long-term health. The researchers aimed to investigate whether maternal Cd exposure during this crucial period of gestation could lead to changes in DNAm patterns that may affect the expression of these imprinted genes. Such alterations in DNAm could subsequently influence various developmental outcomes, including birth weight, which is a vital indicator of neonatal health and can impact a child’s growth and development trajectory. In addition to examining the direct effects of maternal Cd exposure on offspring DNAm and birth weight, the study also sought to evaluate whether certain micronutrients, specifically zinc (Zn) and iron (Fe), could play a protective role in mitigating these adverse outcomes. Both Zn and Fe are essential minerals that have been associated with various physiological processes, including DNA synthesis, repair, and methylation. The hypothesis was that adequate levels of these nutrients might counteract the negative effects of Cd exposure, potentially normalizing DNAm patterns and improving birth weight outcomes [153]. DNAm at regulatory differentially methylated regions (DMRs) of eight imprinted genes was measured in offspring umbilical cord blood (UCB) leukocytes. Elevated maternal Cd levels were associated with lower birth weight (*p* = 0.03). The paternally expressed imprinted gene PEG3, involved in p53/c-myc-mediated processes and critical for brain development [154], showed higher DNAm at its regulatory DMR in association with elevated Cd levels; similar associations were less consistent for IGF2/H19 and MEG3. Cd exposure appeared gene-specific, as DNAm of several other DMRs was not associated with prenatal Cd levels. DNAm at these regulatory DMRs, established during gametogenesis and embryogenesis, may be susceptible to circulating essential metals, including Fe and Zn. Dietary minerals like Zn, Fe, and Ca are known to antagonize Cd absorption [155,156]. Such interventional strategies are active research areas that could inform nutritional policies related to these ubiquitous toxic metals.

Prenatal arsenic (As) exposure has been associated with an increased risk of diseases in adulthood [55]. A study investigated the relationship between maternal drinking water As levels (collected ≤16 weeks’ gestation) and DNAm in UCB from a prospective birth cohort in Bangladesh (n = 44 infants), adjusting for leukocyte-tagged differentially methylated regions. The relationship between As and DNAm at 473,844 CpG sites was analyzed. The median As concentration in water was 12 μg/L (range < 1–510 μg/L). Log10 As was associated with altered DNAm across the epigenome (*p* = 0.002), and As exposure strongly influenced leukocyte distributions in UCB. Specifically, As exposure was linked to altered DNAm and leukocyte subpopulations, particularly CD+4 and CD+8 cells [55]. However, the study did not measure gene expression in cord blood, so it remains unclear whether the DNAm changes corresponded with biological responses.

Current research has highlighted a range of genes and biological pathways that exhibit differential DNA methylation at specific CpG sites in response to prenatal exposure to arsenic (As). This burgeoning body of literature emphasizes the intricate relationship between environmental toxicants and epigenetic modifications that can affect gene expression and developmental outcomes. One study employed an in vivo transplacental model using apolipoprotein E-knockout (ApoE-KO) mice to investigate the impact of arsenic exposure on offspring. This particular model was chosen because it mimics certain metabolic conditions relevant to human health and allows for a more detailed understanding of how prenatal exposures can affect genetic expression in later development stages. In this study, the researchers administered high levels of arsenic to pregnant ApoE-KO mice and subsequently analyzed the liver tissue of their offspring. The results revealed a remarkable increase in the expression of a specific cluster of 51 genes when compared to control mice that had not been exposed to arsenic. Among these genes, HNF4alpha (Hepatocyte Nuclear Factor 4 Alpha) stood out as particularly significant. HNF4alpha is a key transcription factor involved in various metabolic processes, including glucose metabolism and lipid homeostasis, indicating that arsenic exposure during critical developmental windows could have profound implications for the metabolic health of the offspring [157].

The overproduction of reactive oxygen species (ROS) is increasingly recognized as a critical factor in the development of arsenic (As)-related atherosclerosis. Elevated levels of ROS can lead to significant oxidative stress, which disrupts normal cellular processes within the endothelial cells lining blood vessels. This oxidative stress is believed to be a primary driver of endothelial cell proliferation and apoptosis, both of which are pivotal mechanisms in the pathogenesis of atherosclerosis. When ROS levels rise excessively, they can activate various signaling pathways that promote the proliferation of endothelial cells, contributing to the thickening of the vessel wall. Simultaneously, the imbalance caused by increased ROS can trigger programmed cell death, or apoptosis, in endothelial cells. This dual effect can compromise the integrity of the endothelium, making it more susceptible to damage and dysfunction [158].

An in vivo study conducted in rats has provided compelling evidence regarding the impact of chronic exposure to inorganic arsenic (As) on physiological functions, particularly concerning the activity of angiotensin-converting enzyme (ACE) in hepatic tissue. In this study, the rats were subjected to a prolonged exposure period of 200 days, during which they were administered inorganic arsenic at a specific dosage. This extended duration of exposure was designed to mimic the chronic exposure scenarios that can occur in populations living in areas with high arsenic levels in drinking water or through other environmental sources. The results of the study revealed a significant decrease in ACE activity in the liver tissues of the arsenic-exposed rats when compared to the control group. ACE is a critical enzyme involved in the renin–angiotensin system, which plays a vital role in regulating blood pressure, fluid balance, and overall cardiovascular health. The liver is not only a key metabolic organ but also contributes to the synthesis and regulation of various hormones and enzymes, including those involved in blood pressure regulation. The observed reduction in ACE activity may have several implications. First, it suggests that chronic arsenic exposure could disrupt normal hepatic functions, potentially leading to alterations in blood pressure regulation and cardiovascular health. Reduced ACE activity might result in lower levels of angiotensin II, a potent vasoconstrictor, which could affect vascular tone and blood pressure. This dysregulation could contribute to cardiovascular conditions such as hypertension and other related disorders. Moreover, the decrease in ACE activity might also impact the metabolism of other substances in the liver, further complicating the physiological responses to arsenic exposure. For instance, changes in ACE activity could influence the metabolism of hormones and drugs, affecting how the body responds to various treatments and physiological demands [159].

Another study investigated the effects of chronic arsenic (As) exposure on hepatic collagenesis, shedding light on the underlying biological mechanisms involved. The research focused on how long-term exposure to arsenic influences liver fibrosis, a condition characterized by the excessive accumulation of collagen in the liver, which can lead to serious complications such as cirrhosis and liver failure. The findings revealed a strong correlation between chronic arsenic exposure and increased hepatic collagen production, highlighting the potential for arsenic to disrupt normal liver function. Collagen, a structural protein, plays a crucial role in maintaining the integrity of liver tissue. However, its overproduction can result in fibrosis, indicating a pathological response to injury or inflammation. This study emphasizes the significance of understanding how environmental toxins like arsenic contribute to liver pathologies, which could have implications for public health, especially in regions with high arsenic levels in drinking water. Moreover, the study identified a notable relationship between hepatic collagenesis and the expression of key inflammatory cytokines, specifically interleukin 6 (IL-6) and tumor necrosis factor alpha (TNF-α). These cytokines are essential components of the immune response and are known to play critical roles in the hematopoietic cell lineage pathway. IL-6 and TNF-α are involved in various physiological processes, including inflammation, cell proliferation, and differentiation. Their elevated expression in the context of arsenic exposure suggests that arsenic may trigger inflammatory pathways that contribute to collagen deposition in the liver. The connection between chronic arsenic exposure, IL-6, and TNF-α expression underscores the importance of the inflammatory response in the progression of liver fibrosis. Inflammatory cytokines like IL-6 and TNF-α can activate hepatic stellate cells, which are pivotal in the development of fibrosis by producing excessive collagen. This cascade of events illustrates how chronic exposure to environmental toxins can initiate a series of pathological changes that ultimately compromise liver health. This study provides valuable insights into the mechanisms by which chronic arsenic exposure can lead to liver damage and underscores the need for further research to elucidate the pathways involved. Understanding these mechanisms is critical for developing targeted interventions to mitigate the harmful effects of arsenic exposure and prevent liver-related diseases. Additionally, these findings highlight the importance of monitoring environmental arsenic levels and implementing strategies to protect vulnerable populations from exposure to this toxicant [160].

## 6. Discussion

We summarized all the epigenetic modifications caused due to prenatal exposure to heavy metals in Table 2. Further mechanistic studies are needed to elucidate the epigenetic effects of heavy metal exposure. Epigenetic changes that occur during the developmental period often persist even after the environmental stimulus is removed, potentially leading to long-term disorders through sustained epigenetic effects [161,162,163,164,165].

The developing brain is particularly vulnerable to neurotoxicant exposure [167]. The central nervous system in humans continues to develop for several years after birth [139]. Numerous studies have demonstrated that lead (Pb) is toxic to the developing brain across a wide range of exposures [168,169,170]. Recognition of these risks has led to preventative measures, such as the removal of lead additives from gasoline. The causes of neurodevelopmental disorders such as autism, attention deficit disorder, intellectual disability, and cerebral palsy are largely unknown, but these conditions can result in lifelong disabilities [167]. Some industrial chemicals, including Pb, methylmercury (methyl-Hg), and arsenic (As), are known to impair cognitive performance and cause subclinical brain dysfunction in children [167]. Notably, the blood–brain barrier is not fully developed until about six months after birth [171]. As a result, exposure to harmful chemicals during early fetal development can cause brain injury at doses much lower than those affecting adult brain function [167].

While the placenta provides some protection against unwanted chemicals during fetal development, it is not very effective against many environmental pollutants, such as heavy metals [172]. Many metals can readily cross the placenta, and the concentration of mercury (Hg) in umbilical cord blood (UCB) can be significantly higher than in maternal blood [173]. The developmental toxicity of Hg was first identified in the 1960s in Japan, where several infants were born with spasticity, blindness, and intellectual disabilities after their mothers consumed fish from waters contaminated with Hg [167]. The U.S. National Academy of Sciences reviewed several cross-sectional studies and found a strong correlation between fetal neurotoxicity and exposure to low doses of methyl-Hg [174].

These neurodevelopmental disorders contribute to a silent pandemic, hindering societal progress and affecting millions of children worldwide. Impaired brain development and diminished IQ are common consequences, leading to a higher prevalence of intellectual deficits, academic failure, and reduced opportunities for a productive career. For example, most children born between 1960 and 1980 in industrialized countries were exposed to significant amounts of lead (Pb) from gasoline. This exposure likely reduced the number of children with far above-average intelligence (IQ scores above 130) by more than 50% and increased the number of children with IQ scores below 70 [170]. In the United States alone, approximately 100 million children were at risk of airborne Pb exposure during that period. The economic costs associated with this exposure were estimated to range from USD 110 billion to USD 319 billion per birth cohort [175]. Additionally, the approximate cost of prenatal methylmercury (methyl-Hg) toxicity is estimated at USD 8.7 billion annually, with a range of USD 2.2–43.8 billion [176]. These costs primarily arise from reduced economic productivity due to widespread declines in intelligence throughout the exposed individuals’ lifetimes.

Preventing the effects of chemical toxicity from heavy metal exposure is feasible through the implementation of stringent testing regulations. For instance, mandatory laboratory testing of new chemicals before they are marketed can be a highly effective measure to prevent toxicity [177]. Prenatal exposure to mercury (Hg) has been linked to differential DNA methylation (DNAm) at CpG islands in umbilical cord blood (UCB) [96]. Among the top 100 differentially methylated CpGs associated with prenatal Hg levels, two loci (cg27458888 and cg05881762) were found to be hypermethylated and are related to the Ubiquitin Protein Ligase E3A gene (UBE3A). UBE3A is subject to DNAm-dependent genomic imprinting and is associated with Angelman syndrome, a neurodevelopmental disorder characterized by severe intellectual disability [178].

In addition, two loci (cg12419685 and cg17250863) located in CpG islands of the Gamma-Glutamyl Transferase 7 gene (GGT7) were hypermethylated in response to the interaction between Hg and arsenic (As). The GGT7 gene plays a role in the metabolism of glutathione (GSH), an antioxidant that has been shown to protect against methylmercury (MeHg) neurotoxicity in vivo [179,180]. In neurons, glutathione serves as a physiological reservoir of glutamate, a crucial neurotransmitter involved in cognitive processes and previously implicated in MeHg-induced neurotoxicity [181,182]. These findings suggest that even low-dose in utero exposure to Hg and As can have significant epigenetic effects.

## 7. Future Studies

In this review, we cite studies indicating that prenatal exposure to heavy metals correlates with changes in DNA methylation (DNAm) in umbilical cord blood (UCB) across numerous genes. These DNAm changes are associated with a range of short-term and long-term health outcomes in children, such as cardiovascular disease (CVD) and neurodevelopmental issues. However, epidemiological studies are largely correlational, and can be influenced by socioeconomic status, lifestyle factors, genetic makeup, age of the subjects, demographics, and the methods used to measure the association of heavy metal exposure with epigenetic modifications [183]. Therefore, further research using model organisms and embryonic stem cell culture models is necessary to elucidate the underlying mechanisms.

Studies involving mice exposed to toxic metals have shown that heavy metals like lead (Pb), mercury (Hg), and cadmium (Cd) can induce significant neurotoxicity compared to control groups [184]. These studies revealed that key enzymes involved in dopaminergic and serotonergic neurotransmission are affected, leading to a range of neurobehavioral dysfunctions [184]. Additionally, 14-day exposure to Pb and Cd in mice resulted in weight loss, loss of appetite, and increased oxidative stress, indicated by elevated levels of oxidative stress markers such as catalase (CAT) and superoxide dismutase (SOD) [184].

Another study demonstrated that maternal Cd exposure in mice led to increased heart weights in newborn pups and hypertension in adulthood [185]. This exposure significantly altered the levels of essential trace elements in fetal circulation, leading to iron deficiency, which plays a critical role in cardiovascular development [185]. Given that cardiovascular mortality is a leading cause of death and disability globally, environmental risk factors such as heavy metal exposure (e.g., Cd) could be significant contributors [186].

Given the ethical challenges associated with using animal models, stem cells (SCs) have emerged as a promising alternative for studying the cytotoxic effects of environmental stressors like heavy metals [187]. A previous study from our group demonstrated that exposure to physiologically relevant doses of lead (Pb) (0.4–1.9 µM) in human embryonic stem cells (hESCs) downregulated the expression of neural marker genes PAX6 and MSI1. This downregulation was linked to alterations in the DNA methylation status of genes involved in neurogenetic signaling pathways [166]. In mouse embryonic stem cells (mESCs), arsenic (As) was found to inhibit differentiation into cardiomyocytes [188].

Another study screened the effects of three toxic metals—cadmium (Cd), mercury (Hg), and Pb—in a 3D culture system on mouse neural stem cells (mNSCs). The study revealed that all three metals inhibited the differentiation of mNSCs into motor or dopaminergic neurons, with the order of toxicity observed as Pb > Hg > Cd [189]. Cd has also been shown to induce cell death and inhibit the proliferation of adult neural stem cells [190]. Furthermore, a study examining the exposure of mESCs to As, Cd, copper (Cu), Pb, lithium (Li), Hg, and nickel (Ni) found that the levels of methylated H3K27 decreased along with reductions in cell proliferation and expression of DNA repair enzymes [191]. Since umbilical cord blood (UCB) is rich in stem cells, future mechanistic studies examining the impact of heavy metals on these cells could shed light on the epigenetic mechanisms underlying such toxicity.

The protective drugs for heavy metal toxicity include chelating agents and antioxidants [192]. Antioxidants such as melatonin, coenzyme Q10, and quercetin are often used as pretreatment drugs to reduce heavy metal overload. They achieve this by neutralizing cellular reactive oxygen species (ROS) levels and alleviating oxidative stress [192]. Senolytic drugs, which induce apoptosis in senescent cells, are also used as pretreatment drugs. These include BCL family inhibitors, PI3K/AKT inhibitors, FOXO regulators, and resveratrol [193,194,195].

Post-treatment drugs primarily involve chelating agents that bind toxic metal ions and form complexes, facilitating their excretion through the kidneys [196]. The management of heavy metal toxicity is a significant concern in both clinical and environmental health contexts, necessitating the use of effective chelating agents. These agents are compounds that bind to heavy metals in the body, facilitating their excretion and thereby reducing the toxic burden. Among the major chelating agents employed for this purpose are several well-studied and clinically utilized compounds, each with distinct mechanisms of action and applications.
British Anti-Lewisite (BAL) is one of the earliest chelators developed, which was initially used during World War II for treating poisoning from arsenic and other heavy metals. BAL works by forming stable complexes with metals, allowing for their removal from the body.Dimercaptopropane-1-sulfonate (DMPS) is another important chelator that has been used effectively for mercury and lead poisoning. Its water-soluble properties facilitate its use in clinical settings, enabling the rapid mobilization of heavy metals for excretion.Meso-2,3-dimercaptosuccinic acid (DMSA) is widely recognized for its efficacy in treating lead poisoning, especially in children. DMSA is a more favorable option due to its low toxicity and ability to penetrate biological membranes effectively, promoting the elimination of lead and other metals while minimizing adverse effects.Sodium 2,3-monoisoamyl DMSA (MiADMSA) and monomethyl DMSA (MmDMSA) are derivatives of DMSA that have been developed to enhance chelation efficiency and reduce toxicity. These modifications aim to optimize the chelation process while maintaining safety.Monocyclohexyl DMSA (MchDMSA) is another variant designed to improve the pharmacokinetic profile of DMSA, further facilitating heavy metal removal from the body.Calcium disodium ethylenediamine tetra-acetic acid (CaNa2EDTA) is a well-established chelator used primarily for lead poisoning. EDTA binds to lead ions and other metals, enabling their renal excretion. The use of calcium disodium EDTA is particularly advantageous in pediatric populations due to its safety profile.Calcium trisodium diethylenetriaminepentaacetate is another chelator similar to EDTA that is used for various heavy metals, including lead and cadmium. It offers similar benefits in terms of efficacy and safety.D-penicillamine is an alternative chelating agent originally derived from penicillin, which is commonly used to treat copper overload in conditions like Wilson’s disease. It also has applications in the removal of lead and mercury, though its use can be limited by side effects.Tetraethylenetetramine (TETA), also known as trientine, serves as a chelator primarily for copper and is often utilized in the management of Wilson’s disease.Nitrilotriacetic acid (NTA) is a synthetic chelating agent that has been studied for its ability to bind various metals and has applications in both industrial and clinical settings.Deferoxamine (DFO) is a crucial chelator specifically for iron overload conditions, such as thalassemia and hemochromatosis. It forms complexes with excess iron, facilitating its elimination from the body and reducing the risk of iron-induced toxicity.Deferiprone (L1) is another iron chelator that has been effective in managing iron overload. It is particularly useful in patients who do not respond well to DFO and can help maintain lower iron levels in the body.

The landscape of chelating agents available for treating heavy metal toxicity is diverse, with each agent offering unique benefits and limitations. The choice of a specific chelator depends on various factors, including the type of metal involved, the severity of toxicity, patient characteristics, and the desired therapeutic outcome. Ongoing research and clinical trials continue to refine these agents, enhancing their efficacy and safety profiles while addressing the pressing issue of heavy metal exposure in vulnerable populations [192].

Free radical scavenging antioxidants, including vitamins E and C, as well as astaxanthin, have shown considerable potential as therapeutic agents in mitigating oxidative stress induced by heavy metal toxicity. Oxidative stress arises when there is an imbalance between the production of reactive oxygen species (ROS) and the body’s ability to neutralize these harmful molecules through its antioxidant defenses. Heavy metals such as lead (Pb), mercury (Hg), cadmium (Cd), and arsenic (As) exacerbate this imbalance by promoting excessive ROS generation, leading to cellular damage, inflammation, and disruption of vital biological processes.
Vitamin E (α-tocopherol) is a lipid-soluble antioxidant that plays a critical role in protecting cell membranes from oxidative damage. It works by scavenging lipid peroxyl radicals, thereby inhibiting lipid peroxidation, a process that can lead to cell membrane breakdown, organelle dysfunction, and cell death. In the context of heavy metal toxicity, vitamin E has been shown to mitigate oxidative damage in organs such as the liver, kidneys, and brain. For instance, studies have demonstrated that vitamin E supplementation can reduce lipid peroxidation and improve antioxidant enzyme activities in animals exposed to cadmium, suggesting its protective role against Cd-induced hepatotoxicity and nephrotoxicity. Moreover, vitamin E may reduce lead-induced neurotoxicity by preventing oxidative damage to neural cells, offering protection against cognitive and developmental impairments associated with early-life lead exposure.Vitamin C (ascorbic acid), a water-soluble antioxidant, also plays a crucial role in combating oxidative stress induced by heavy metal exposure. It acts as a direct scavenger of ROS, neutralizing free radicals before they can cause cellular harm. Vitamin C also enhances the activity of other antioxidants, including vitamin E, by regenerating its active form after it has neutralized free radicals. Research suggests that vitamin C can alleviate the toxic effects of heavy metals like lead and mercury by reducing ROS levels, protecting against DNA damage, and enhancing the excretion of these metals from the body. In individuals exposed to lead, for example, vitamin C has been shown to reduce blood lead levels and improve antioxidant status, making it a valuable component of chelation therapy. Additionally, vitamin C can ameliorate mercury-induced oxidative damage in the kidneys and liver by strengthening endogenous antioxidant defenses.Astaxanthin, a naturally occurring carotenoid found in marine organisms such as microalgae, salmon, and shrimp, has emerged as a powerful antioxidant with potential therapeutic benefits in heavy metal toxicity. Astaxanthin is known for its superior free radical scavenging capabilities, which are significantly stronger than those of vitamins E and C. Its unique molecular structure allows it to span cell membranes and protect both the lipid bilayer and aqueous compartments from oxidative damage. In studies examining the protective effects of astaxanthin against cadmium and lead toxicity, it has been shown to reduce oxidative stress markers, enhance the activity of antioxidant enzymes, and improve mitochondrial function. Additionally, astaxanthin’s anti-inflammatory properties further contribute to its protective effects by mitigating the inflammatory responses triggered by heavy metal exposure, which are often driven by oxidative stress.

These antioxidants not only protect cells and tissues from oxidative damage but may also interact with heavy metals to enhance their detoxification and removal from the body. For instance, both vitamins E and C have been shown to enhance the excretion of metals like lead and mercury, working synergistically with chelation therapies to reduce the toxic burden. The ability of antioxidants to reduce oxidative stress also helps to preserve the function of critical detoxification organs such as the liver and kidneys, which are often compromised in cases of heavy metal poisoning. Furthermore, the neuroprotective effects of these antioxidants are of particular interest, given the vulnerability of the central nervous system to oxidative damage caused by heavy metals. Lead, mercury, and arsenic are well-known neurotoxins that can impair cognitive development in children and contribute to neurodegenerative diseases in adults. Antioxidants like vitamins E and C, as well as astaxanthin, have been shown to protect neurons from oxidative damage, potentially mitigating the neurodevelopmental and neurodegenerative effects of early-life and chronic heavy metal exposure. In conclusion, free radical scavenging antioxidants such as vitamins E and C, along with astaxanthin, offer promising avenues for reducing the harmful effects of oxidative stress induced by heavy metal toxicity. Their ability to neutralize ROS, protect cellular structures, enhance the excretion of toxic metals, and support the body’s natural antioxidant defenses positions them as valuable components in both preventive and therapeutic strategies against heavy metal exposure. Future research should continue to explore the optimal dosages, combinations, and long-term benefits of these antioxidants in populations at risk of heavy metal toxicity, with an emphasis on vulnerable groups such as pregnant women, children, and individuals living in heavily polluted environments [197,198]. Further research is needed to develop additional therapeutic agents for combatting heavy metal toxicity.

## 8. Conclusions

This review has highlighted the use of DNA methylation (DNAm) in umbilical cord blood (UCB) as an epigenetic biomarker for assessing heavy metal exposure and its correlations with childhood health outcomes, particularly neurodevelopment. We emphasized the necessity for further animal studies and research on mediators of heavy metal toxicity. Future research should concentrate on examining individual cell types in UCB and their spatial–temporal arrangement, employing methods such as single-cell and spatial transcriptomics. There is also a need to explore other epigenetic mechanisms in UCB beyond DNAm. Although a few studies have investigated DNA hydroxymethylation (DNAhm), more research in this area is needed. Additional epigenetic mechanisms that warrant exploration include histone modifications, long noncoding RNAs, and epitranscriptomics under the study of RNA modifications. The development of new technologies continues to advance, enabling the investigation of these approaches at the single-cell and spatial–temporal levels in UCB tissues.

## Figures and Tables

**Figure 1 cells-13-01775-f001:**
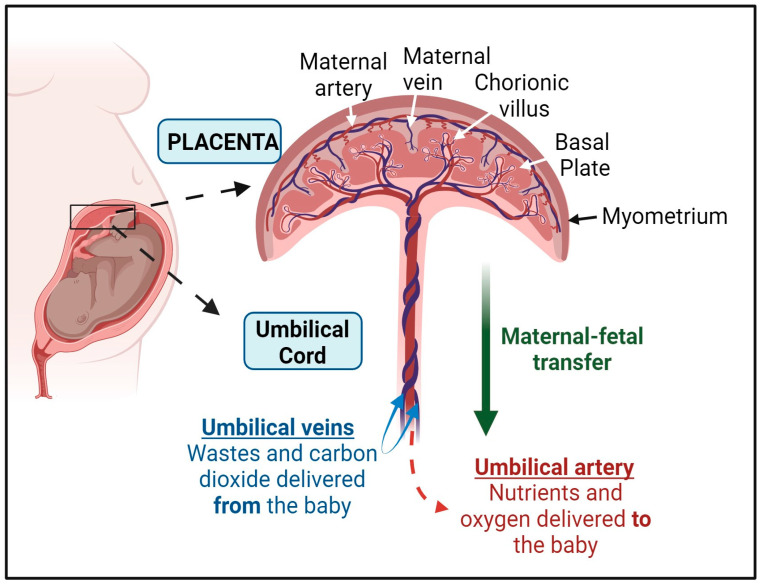
A growing fetus inside the mother’s uterus showing the umbilical cord and placenta. The structure of the human placenta showing the direction of maternal–fetal transfer. The fetal side of the placenta comprises umbilical arteries (which deliver nutrients and oxygen to the growing fetus) and umbilical veins (which transport back waste products and CO_2_ from the fetus to the maternal circulation). The maternal side consists of a basal plate with maternal vessels. The middle portion of the placenta consists of chorionic villi.

**Figure 2 cells-13-01775-f002:**
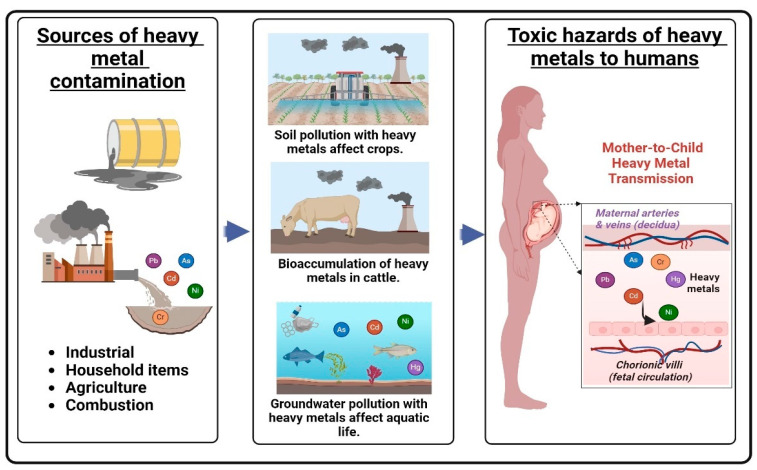
Heavy metal transition from mother to fetus. Heavy metals are released in the environment from several sources, such as mining, oil refineries, combustion by-products, fertilizers, and pesticides. Heavy metals can get easily transmitted via the maternal circulation and affect the growing fetus.

**Table 1 cells-13-01775-t001:** Studies showing null association of prenatal heavy metal exposure with epigenetic changes in cord blood.

Heavy Metal	Country	Phenotypic Outcome	Tissue Methylation Measured	Reference
Arsenic (As)	Norway	No association observed between maternal As levels and 5 mC in pregnant mothers or newborns	Cord blood and maternal whole blood	[65]
Lead (Pb)	Mexico	No associations were reported between cord blood Pb and cord blood genomic DNA methylation	Leukocyte DNA from umbilical cord blood samples	[66]
		No association of prenatal Pb exposures with telomere length and DNA-methylation-based predictors of age in cord blood	Whole cord blood	[67]
Mercury (Hg)	USA	No association reported between prenatal maternal RBC-Hg and 5-methylcytosine (%-5mC)	Whole cord blood	[68]
	Faroe Islands, Denmark	Most of differentially methylated CpG sites (214 sites) were not associated with MeHg	Total cord blood	[69]
Iron	Several European countries	Cord blood serum ferritin concentrations were not associated with cord blood DNA methylation levels at three identified CpGs: two CpGs (cg02806645 and cg06322988) in *PRR23A* and one CpG (cg04468817) in *PRSS22*	Whole cord blood	[70]

**Table 2 cells-13-01775-t002:** Epigenetic modifications associated with prenatal exposure to heavy metals.

**Heavy Metal**	**Country**	**Key Attributes**	**Key Epigenetic** **Outcome**	**Reference**
Arsenic (As)	Bangladesh	Cancer-related genes	Decrease in DNA methylation in boys	[82]
	Bangladesh	DNA methylation in cord blood adjusting for leukocyte-tagged differentially methylated regions	Significantly associated with leukocyte subpopulations, specifically CD+4 and CD+8 populations	[55]
Mercury (Hg)	Japan	HDHD1 gene that encodes pseudouridine-5′-phosphatase (PUDP)	DNA methylation within HDHD1 increased with Hg concentration	[83]
	Korea	Hg levels were measured in cord blood to elucidate the association between prenatal and early childhood Hg exposure and autistic behaviors in preschool-age children	Hg concentrations at late pregnancy, in cord blood, and at 2 and 3 years of age were associated with autistic behaviors at 5 years of age	[139]
	Croatia and Italy	Hg from fish consumption during pregnancy on newborn’s brain development and child neurodevelopment using UCB	Morphological changes in the brains of newborns were detected on Hg exposure. The width of the frontal gyrus was greater in the exposed group and the length of the cerebellum narrower than in the unexposed group	[141]
	USA	Global 5-hydroxymethylcytosine (%-5hmC) and 5-methylcytosine(%-5mC) DNA content in blood in preterm birth cohort	Hg exposure was associated with lower global %-5hmC DNA content in cord blood	[68]
	Spain, Korea, USA, Japan, UK, Norway, Greece	Differential DNAm at cg24184221 in *MED31* (gene involved in lipid metabolism and RNA Pol II function) in relation to prenatal MeHg exposure	Hypo/hypermethylation of several genes involved in growth and cell cycle processes during fetal development	[98]
	USA	DNA methylation changes in a genomic region of the Paraoxonase 1 (PON1) gene	Higher DNA methylation levels of the PON1 region were associated with lower cognitive test scores in early childhood for both sexes	[99]
As and Hg co-exposure	USA	There is a high proportion of loci located in CpG islands and in south shore regions and all these loci were hypermethylated	A decrease in the proportion of monocytes and an increase in B-cell population in female infants	[96]
	Tanzania	Genes affecting placental function, oxidative stress, and fetal growth	As and Hg exposure was related to an increased incidence of spontaneous abortion, stillbirth/preterm birth, etc.	[100]
**Heavy Metal**	**Country**	**Gene/Element**	**Epigenetic Outcome**	**Reference**
Cadmium (Cd)	USA	Genes that encode for proteins that control transcriptional regulation and apoptosis	Differentially methylated genes showed hypermethylation	[115]
	Korea	Two differentially methylated CpG sites, cg05537752 and cg24904393	Fetus susceptible to Cd-induced epigenetic modifications (trimester specific)	[116]
	Bangladesh	CpG sites that were positively associated with Cd were inversely correlated with birth weight	In girls, methylation changes in genes related to organ development, morphology, and mineralization of bone; in boys it was cell death-related genes	[117]
	Bangladesh	Six differentially methylated CpG sites (DMPs) in the children’s PBMCs were associated with gestational Cd exposure and 11 DMPs with the children’s long-term Cd exposure	Gestational Cd exposure leads to hypomethylation of DMRs and adverse outcomes in children	[118]
	USA	In newborn cord blood and maternal blood, 641 and 1,945 Cd-associated DMRs were identified, respectively	The top three functional categories for genes affected were body mass index (BMI), blood pressure, and body weight	[119]
	Korea	A decrease in gestational age was observed by DNA methylation at a specific CpG site, cg21010642	The CpG site was annotated to a gene involved in early embryonic development, thus causing preterm birth	[120]
	USA	Elevated maternal Cd levels were associated with higher DNA methylation at the DMR regulating PEG3, and less consistently at IGF2/H19 and MEG3	Elevated maternal blood Cd levels were associated with lower birth weight	[153]
Lead	USA	Mothers with high neonatal blood Pb levels correlate with altered DNA methylation at 564 loci in their children’s neonatal blood	Memory loss and loss of cognitive function	[131]
	Mexico	Several 5hmC and 5mC clusters as potential candidates for sex-specific epigenetic biomarkers for prenatal Pb exposure	Genes for neurodevelopment	[166]
	Mexico	An inverse dose–response relationship was observed of patella Pb with UCB methylation	The epigenome of the developing fetus is influenced by maternal cumulative Pb burdens, leading to disease susceptibility throughout life course	[66]
	USA	Pb concentrations in RBCs from prenatal maternal blood samples were measured and genome-wide methylation levels at 482,397 CpG loci in UCB were analyzed	In utero Pb exposure gave rise to altered methylation levels at CpGs located within or near important regulatory genes of hematopoietic and nervous functions (i.e., CLEC11A and DNHD1)	[142]
Heavy metal co-exposure	Japan	Prenatal exposure—As, Cd, Hg, Pb, antimony (Sb)—with global DNA methylation in UCB DNA were determined	Significant positive correlations were observed among Pb levels, maternal age, and hmC content; consistent positive associations between Pb and Sb levels and mC and hmC content were observed; these are biomarkers for future disease risks	[133]
	China	Pb, Cd, Mn, and Cr on neonatal DNA methylation patterns	Genes involved in neurodevelopment were affected	[134]
	China	A panel of 16 metals and 3 oxidative stress biomarkers were measured; Alu and LINE-in cord blood were analyzed	Positive associations were observed between As, Cd, Tl, Ba, Ni, V, Co, Zn, Cu, Se, and Mo and at least one oxidative stress biomarker	[135]
	USA		Prenatal metal exposure was associated with DNAm, including DMRs annotated to genes involved in neurodevelopment	[136]
	Mexico	The authors investigated the impact of prenatal metal exposure (As, Cu, Hg, Mn, Mo, Pb, Se, and Zn) on DNA methylation and mRNA expression in 181 umbilical cord blood samples	DNA methylation is affected by prenatal metal exposure, which could cause alterations in the expression of repair genes, leading to lower capacity for DNA damage repair in newborns	[137]

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
