# Peer review of "Heavy Metals in Umbilical Cord Blood: Effects on Epigenetics and Child Development"

_cells, 2024, doi:10.3390/cells13211775_

Round 1
Reviewer 1 Report
Comments and Suggestions for Authors
The manuscript reviews the effects of heavy metals in umbilical cord blood on epigenetics and child development, the topic is of interest and I would recommend that the authors address the following issues when revising the manuscript:
1. In lines 37-44, I would recommend that the authors not to write Part 1, 2, 3, and 4.
2. The author omitted to write Table 1 in the main text.
3. I recommend that the author replace the subheading ‘6. Discussion’, that is, do not use Discussion as a subheading.
4. The use of acronyms in this manuscript is not standardized.
Comments on the Quality of English Language
Please double-check the whole text to eliminate some writing errors.
Author Response
Reviewer 1:
Comments and Suggestions for Authors
The manuscript reviews the effects of heavy metals in umbilical cord blood on epigenetics and child development, the topic is of interest and I would recommend that the authors address the following issues when revising the manuscript:
- In lines 37-44, I would recommend that the authors not to write Part 1, 2, 3, and 4.
Authors’ response : We have deleted Parts 1, 2, 3 , 4.
- The author omitted to write Table 1 in the main text.
Authors’ response : We apologize for the oversight. We have mentioned Table 1 in lines 607-608.
- I recommend that the author replace the subheading ‘6. Discussion’, that is, do not use Discussion as a subheading.
Authors’ response : We have deleted number 6 from Discussion.
- The use of acronyms in this manuscript is not standardized.
Authors’ response : We thank the reviewer for this very valid point. We had to use a lot of non-standard acronyms in the manuscript to bypass the plagiarism and overlap check software which identifies commonly used acronyms as overlap.
Comments on the Quality of English Language
Please double-check the whole text to eliminate some writing errors.
Authors’ response : We have once again checked the manuscript for any grammatical or typological errors.
Reviewer 2:
Comments and Suggestions for Authors
The research primarily investigates the impact of heavy metal exposure during pregnancy on epigenetic modifications (specifically DNA methylation) in newborns, as evidenced by changes in their umbilical cord blood. The authors aim to understand how these epigenetic changes may influence child development and long-term health.
While the topic of heavy metal exposure and its effects on child health is not entirely novel, the focus on epigenetic mechanisms and the use of umbilical cord blood as a biomarker for early-life exposure adds a layer of originality and relevance to the field. The research addresses a specific gap by providing insights into the potential long-term consequences of prenatal heavy metal exposure at a molecular level.
This review contributes to the existing studies by:
- Highlighting the importance of epigenetic mechanisms in mediating the effects of heavy metals on child development.
- Emphasizing the role of the placenta as a protective barrier and a valuable source of information for studying prenatal exposures.
- Providing a comprehensive overview of the research on heavy metal exposure and epigenetic changes in umbilical cord blood.
Authors’ response : The authors would like to appreciate the reviewers for this very encouraging feedback.
1.While the review provides a solid foundation for understanding the topic, it would benefit from a more in-depth discussion of the methodologies employed in the studies cited. The authors could consider:
- a) Discussing the limitations of different methods used to measure heavy metal exposure and DNA methylation.
Authors’ response : We appreciate the reviewer for pointing this very valid point. However, in this article we wanted to primarily focus how the different heavy metals ( e.g: arsenic, mercury, cadmium, and lead) which are common environmental contaminants affect the developing fetus by causing short term and long-term health disorders via various epigenetic changes. Elaborating on the methodologies employed in those studies is beyond the scope of this review.
- b) Examining the potential confounding factors that could influence the observed associations between heavy metal exposure and epigenetic changes.
Authors’ response: We have discussed this in page 15 , lines 689-692.
- The conclusions drawn in the review are generally consistent with the evidence presented. However, the authors could strengthen their argument by:
a)Providing a more nuanced interpretation of the findings, considering the potential variability in individual responses to heavy metal exposure.
Authors’ response: We would foremost like to thank the reviewer for highlighting a very important point. Genetic makeup, lifestyle factors , demographics indeed matter how a particular individual would respond to heavy metal exposure. To portray a more global picture, we have discussed the studies in this manuscript by mentioning the country where the epidemiological study was conducted for each specific case.
b)Discussing the implications of the observed epigenetic changes for child health and development in greater detail.
Authors’ response: We have conducted a thorough recent literature search and provided several examples of health defects in the section “Short-term and long-term health outcome of fetal exposures to heavy metals in relation DNA methylation”. We have discussed health deficits such as autism, neurodevelopmental deficits, bone defects, heart damage, birth weight etc. caused due to prenatal heavy metal exposure. However, it is beyond the scope of the review to go into further details.
3.The references appear to be appropriate and relevant to the topic. However, the authors could consider providing a more critical evaluation of the cited literature, highlighting both the strengths and weaknesses of different studies.
Authors’ response : We sincerely appreciate the reviewer for the valuable input. In this review , as mentioned in the section” Search Strategy and Selection Criteria” we have not discussed the null findings and elaborated on the studies which discussed a phenotypic outcome related to heavy metal exposure. Results for epidemiological studies vary due to differences in populations, methods used, measurement, environmental factors etc. and therefore going in-depth into the strength and weakness of every individual study is beyond the scope of the review.
- Overall, this review provides a valuable contribution to the field of environmental health and child development. By addressing the impact of heavy metal exposure on epigenetics, the authors offer new insights into the potential long-term consequences of prenatal exposures. However, the review could be strengthened by a more in-depth discussion of the methodologies and a more nuanced interpretation of the findings.
Authors’ response : We once again sincerely thank the reviewer for the positive feedback. However, keeping in mind the overall focus and objective of our review we have excluded the methodologies and variation in the individual responses to heavy metal exposure in this manuscript.
Reviewer 3
Comments and Suggestions for Authors
1)The manuscript text is clearly written and well-organized. The topic is of importance.
Authors’ response : The authors would like to thank the reviewer for the very encouraging comment.
2) My major concerns relate to the methods used for the literature review and the presentation of the literature review. While the authors do not claim to be following the methods of a systematic literature review, we still need to know how the identified the body of literature: search source, time period, language, etc. Do the authors summarize all papers not showing associations? Is there a summary of the weaknesses of the literature on metals and cord blood DNAm generally or by metal?
Authors’ response: We apologize for the oversight. We have included a section called “ Search Strategy and Selection Criteria” in page 5, lines 161-173.
3) My second concern is related to the first. The manuscript needs a table to illustrate the literature being reviewed with columns on key issues. For example, sex specific effects are important and not always presented. Where the study was done could be important. There needs to be systematic information about what is being shared with the readers.
Authors’ response: We appreciate the reviewer for this critical point. We have highlighted in Table 1 (Columnïƒ Phenotypic Outcome – the sex where the phenotypic outcome was more pronounced wherever applicable).

Reviewer 2 Report
Comments and Suggestions for Authors
The research primarily investigates the impact of heavy metal exposure during pregnancy on epigenetic modifications (specifically DNA methylation) in newborns, as evidenced by changes in their umbilical cord blood. The authors aim to understand how these epigenetic changes may influence child development and long-term health.
While the topic of heavy metal exposure and its effects on child health is not entirely novel, the focus on epigenetic mechanisms and the use of umbilical cord blood as a biomarker for early-life exposure adds a layer of originality and relevance to the field. The research addresses a specific gap by providing insights into the potential long-term consequences of prenatal heavy metal exposure at a molecular level.
This review contributes to the existing studies by:
- Highlighting the importance of epigenetic mechanisms in mediating the effects of heavy metals on child development.
- Emphasizing the role of the placenta as a protective barrier and a valuable source of information for studying prenatal exposures.
- Providing a comprehensive overview of the research on heavy metal exposure and epigenetic changes in umbilical cord blood.
While the review provides a solid foundation for understanding the topic, it would benefit from a more in-depth discussion of the methodologies employed in the studies cited. The authors could consider:
- Discussing the limitations of different methods used to measure heavy metal exposure and DNA methylation.
- Examining the potential confounding factors that could influence the observed associations between heavy metal exposure and epigenetic changes.
The conclusions drawn in the review are generally consistent with the evidence presented. However, the authors could strengthen their argument by:
- Providing a more nuanced interpretation of the findings, considering the potential variability in individual responses to heavy metal exposure.
- Discussing the implications of the observed epigenetic changes for child health and development in greater detail.
The references appear to be appropriate and relevant to the topic. However, the authors could consider providing a more critical evaluation of the cited literature, highlighting both the strengths and weaknesses of different studies.
Overall, this review provides a valuable contribution to the field of environmental health and child development. By addressing the impact of heavy metal exposure on epigenetics, the authors offer new insights into the potential long-term consequences of prenatal exposures. However, the review could be strengthened by a more in-depth discussion of the methodologies and a more nuanced interpretation of the findings.
Author Response

(The authors gave the same response as above.)

Reviewer 3 Report
Comments and Suggestions for Authors
The manuscript text is clearly written and well-organized. The topic is of importance.
My major concerns relate to the methods used for the literature review and the presentation of the literature review. While the authors do not claim to be following the methods of a systematic literature review, we still need to know how the identified the body of literature: search source, time period, language, etc. Do the authors summarize all papers not showing associations? Is there a summary of the weaknesses of the literature on metals and cord blood DNAm generally or by metal?
My second concern is related to the first. The manuscript needs a table to illustrate the literature being reviewed with columns on key issues. For example, sex specific effects are important and not always presented. Where the study was done could be important. There needs to be systematic information about what is being shared with the readers.
Author Response

(The authors gave the same response as above.)

Round 2
Reviewer 3 Report
Comments and Suggestions for Authors
1. The authors have chosen to exclude literature with null associations. That is unacceptable. How can you assess a topic and only review the publications with significant associations? You must include all publications on the topics to provide a balanced summary.
2. The column added to describe the results is simplistic. The authors need to think about the key attributes of the publications that should be described for readers as well as the key outcomes.
Author Response
1.The authors have chosen to exclude literature with null associations. That is unacceptable. How can you assess a topic and only review the publications with significant associations? You must include all publications on the topics to provide a balanced summary.
Author’s response: We apologize for the oversight and have included Table2 with the studies showing null associations.
2.The column added to describe the results is simplistic. The authors need to think about the key attributes of the publications that should be described for readers as well as the key outcomes.
Author’s response: We would like to thank the reviewer for highlighting this very important point. The studies indicated in the table have been described in detail under each section of heavy metal. The table was meant to provide the readers with a synopsis of the major outcomes for each metal. According to the suggestion of the recommender, we have modified the column name from Gene/ Element to Key attribute and the column name from Epigenetic outcome to Key epigenetic outcome to highlight the major phenotypic changes occurring due to the prenatal heavy metal exposure.